# Evaluation of larvicidal potential against larvae of *Aedes aegypti* (Linnaeus, 1762) and of the antimicrobial activity of essential oil obtained from the leaves of *Origanum majorana* L.

**Renata do Socorro Barbosa Chaves**[1], **Rosany Lopes Martins**[1], **Alex Bruno Lobato Rodrigues**[1], **Érica de Menezes Rabelo**[1], **Ana Luzia Ferreira Farias**[1], **Lethicia Barreto Brandão**[1], **Lizandra Lima Santos**[1], **Allan Kardec Ribeiro Galardo**[2], **Sheylla Susan Moreira da Silva de Almeida**[1]*

1 Laboratory of Pharmacognosy and Phytochemistry, Federal University of Amapá, Macapá, Amapá, Brasil, 2 Laboratory of Entomology Medical of Institute of Scientific and Technological Research of the State of Amapá (IEPA), Macapá, Amapá, Brasil

* sheyllasusan@yahoo.com.br

## Abstract

This study evaluated the larvicidal activity of *Origanum majorana* Linnaeus essential oil, identified the chemical composition, evaluated the antimicrobial, cytotoxic and antioxidant potential. The larvicidal activity was evaluated against larvae of the third stage of *Aedes aegypti* Linaeus, whereas the chemical composition was identified by gas chromatography coupled to mass spectrometer, the antimicrobial activity was carried out against the bacteria *Pseudomonas aeruginosa*, *Escherichia coli* and *Staphylococcus auereus*, the antioxidant activity was evaluated from of 2.2-diphenyl-1-picryl-hydrazila sequestration and *Artemia salina* Leach cytotoxicity. Regarding to the results, the larvicidal activity showed that *O. majorana* L. essential oil caused high mortality in *A. aegypti* L. larvae. In the chromatographic analysis, the main component found in *O. majorana* L. essential oil was pulegone (57.05%), followed by the other components verbenone (16.92%), trans-p-menthan-2-one (8.57%), iso-menthone (5.58%), piperitone (2.83%), 3-octanol (2.35%) and isopulegol (1.47%). The antimicrobial activity showed that *E. coli* and *P. aeruginosa* bacteria were more sensitive to oil than *S. aureus*, which was resistant at all concentrations. Essential oil did not present antioxidant activity, but it has high cytotoxic activity against *A. salina* L.

## Introduction

Dengue remains an important public health problem in Brazil, even after the introduction and recent dissemination of the Zika and chikungunya viruses [1,2]. The disease presents a great epidemic potential, affecting all regions of Brazil [3,4].

The *Aedes aegypti* Linnaeus [5] mosquito is a vector of viruses that cause diseases known as dengue, chikungunya, and zika [6]. It has holometabolic development, with egg, larva, pupa

**Data Availability Statement:** All relevant data are within the manuscript and its Supporting Information files.

**Funding:** Amapá Foundation for Research Support (FAPEAP). To the Research Program of SUS - PPSUS - Ministry of Health - Responsible for financing the research project. Coordination of Improvement of Higher Education Personnel (CAPES) / Ministry of Education (MEC) and to the National Council of Scientific and Technological Development - CNPQ - for the promotion of scholarships to students participating in the project.

**Competing interests:** The authors have declared that no competing interests exist.

and adult phases. Because it is a mosquito highly adapted to the urban environment, its most common breeding sites are artificial containers that accumulate water, such as bottles, tires, cans, and pots [7].

Among the control policies adopted in Brazil, the mechanical control is carried out by ACE (Agents to Combat Endemics), with the participation of the population, aiming at the protection, destruction or adequate allocation of potent breeding sites. The intensive collaboration of the population is crucial to hinder the proliferation and installation of the mosquito. In addition, it reinforces the need for adequate sanitary conditions in the cities, eliminating stocks of water that allow eggs to hatch. An important strategy is the promotion of educational actions during home visits made regularly by the health agents [8].

The spread and flow of various serotypes of the dengue virus over the years also have a significant influence on epidemics, as well as an increase in cases diagnosed for the most severe form of the disease. These factors demonstrate the importance of introducing preventive measures in order to reduce dengue rates [9].

To combat these disease vectors, insecticides are the most used products, but they have several disadvantages, as they can be the source of several environmental problems [10, 11]. In fact, most of the chemical insecticides used cause a major problem, especially the development of mosquito resistance [12, 13].

In addition, the literature reports that *Origanum majorana* species have attracted consumers' attention due to their antimicrobial, antifungal, insecticidal and antioxidant effects on human health [14]. The crude extract of *O. majorana* and essential oil show significant results in inhibiting the growth of bacteria and fungi and the synthesis of microbial metabolites [15–17].

*Origanum majorana* Linnaeus [18] belongs to the Lamiaceae [19] family, and it contains several terpenoids, which are isolated from aerial parts of the *Origanum* plant and exhibit antimicrobial, antiviral and antioxidant properties, without toxic effects [20, 21].

Previous studies have reported the potential use of the *O. majorana* ethanolic extract as an anticancer agent [22, 23], while the tea extract has been shown to have immunostimulating, antigenotoxic and antimutagenic properties [24, 25]. These activities are attributed to the chemical composition, characterized as rich in flavonoids and terpenoids [26].

The antioxidant activity of essential oils through the elimination of free radicals and inhibition of oxidation of linoleic acid, can be useful for the food industry to prolong the stability of food storage [27].

Thus, the search for natural antimicrobial compounds has been of great interest at the scientific and clinical level, so that aromatic and medicinal plants play a central role in the search for biologically active molecules, against different microorganisms [28].

The antimicrobial and antioxidant properties of many spices and their essential oils have been known for a long time, but only in recent years have consumers given proper attention to the use of these substances [29]. Because many plants are toxic to mosquitoes, the mixture of essential oils may represent an efficient outlet for this problem, compared to the *A. aegypti* L. mosquito [30].

In the literature, there are no reports on larvicidal activity against *A. aegypti* L. and cytotoxicity against *A. salina* L. and few studies have been reported on the antioxidant and antimicrobial effects of the essential oils of this species.

The active effects of the *O. majorana* species are not as studied and especially the larvicidal effect. So far, we have not found such studies related to this species with unique chemical composition in the country. Thus, in this article a first study was carried out on the larvicidal activity of essential oil.

It is in this sense that the work was carried out for the first time in Macapá and aims to study the larvicidal activity of the essential oils of *O. majorana* (Lamiaceae), cultivated in the north of the state against the larvae of *A. aegypti*, the vector of the dengue virus, Zika, chikungunya and malaria. In addition, to determine the chemical composition, to evaluate the antimicrobial activity against *E. coli*, *P. aeruginosa* and *S. aureus* bacteria, to determine the antioxidant potential through the sequestration of DPPH and cytotoxicity against *A. salina* L. of *O. majorana* L. essential oil.

## Materials and methods

### Plant material

The leaves of *O. majorana* L. were collected in the district of Fazendinha (00 "36'955" S and 51 "11'03'77" W) in the Municipality of Macapá, Amapá. Five samples of the plant species were deposited at the Amapaense Herbarium (HAMAB) of the Institute of Scientific Research and Technology of Amapá (IEPA).

### Essential oil obtaining

The hydrodistillation process using the Clevenger type apparatus, 131 g of *O. majorana* L. dried leaves were dried at 45 ˚C for a period of 2 h [31] obtained the essential oil (EO). The EO was kept under refrigeration (4 ˚C).

### Identification of the chemical composition by gas chromatography coupled to mass spectrometer (GC-MS)

The EO analysis was performed by Gas Chromatography coupled to the Mass Spectrometer (GC-MS) of the Museu Paraense Emílio Goeldi. The Shimadzu equipment, model GCMS-QP 5000 A was used. A fused silica capillary column (OPTIMA®-5-0.25 μm) was used. It has 30 m of length and 0.25 mm of internal diameter and nitrogen as carrier gas. The operating conditions of the gas chromatograph were: internal column pressure 67.5 kPa, division ratio 1:20, gas flow at column 1.2 mL.min$^{-1}$ (210 ˚C), injector temperature 260 ˚C, temperature detector or interface of 280 ˚C. The initial column temperature was 50 ˚C, followed by an increase from 6 ˚C.min$^{-1}$ to 260 ˚C kept constant for 30 min. The mass spectrometer was programmed to perform readings at intervals of 29–400 Da, at intervals of 0.5 s with ionization energy of 70 eV. 1 μL of each sample with a concentration of 10.000 ppm dissolved in hexane was injected.

The identification of the chemical compounds present in the EO was made from the comparisons of the Indices of Retention (IR) and Kovats (IK) of the homologous series of n-alkanes (C8-C26) and the literature [32]. Identification was also made by combining the spectra obtained by the analysis performed on the Lab solutions GC-MS version 2.50 Sigma–Aldrich, St. Louis, MO, USA e software equipment of the mass spectra of the NIST05 and WILEY'S libraries.

### Larvicidal activity against *A. aegypti* L. larvae

The data from this study were interpreted to identify compounds that showed toxicity of the essential oil of *O. majorana*. The identification of the active substances against *A. aegypti* larvae helped to interpret the toxicity of the essential oil.

The *A. aegypti* L. larvae used in the bioassay came from the colony (strain Rockfeller) kept in the Medical Entomology Laboratory of the Institute of Scientific and Technological Research of the State of Amapá (IEPA). The methodology used followed the World Health Organization standard protocol [33] with adaptations.

The procedure started with the separation of 18 beakers of 50 mL and in each Becker, there were added 25 larvae of the third instar of *A. aegypti* L. Then they were reserved in a room with conditions of ambient temperature between 25 to 30 ˚C and photoperiod of 12 h.

Preparation of the samples started after 24 h. The stock solution was prepared with 4.5 mL of Tween 80, 85.5 mL of distilled water and 0.09 g of the EO sample of *O. majorana* L. The positive control was prepared with 17.5 mL of Tween 80 dissolved in 350 mL of distilled water, and the larvicidal esbiothrin as the positive control.

After the preliminary tests, the aqueous solution was diluted in the following concentrations: 100, 80, 60, 40, 20, 10, and 1 µg.mL$^{-1}$. Each concentration was tested in triplicate, and 25 larvae of the *A. aegypti* L. mosquito in the 3rd young stage (L3) were used. They were pipetted into a 100 mL beaker containing distilled water, then they were transferred into the test vessels, minimizing the time between the preparation of the first and last samples. During the experiment, the average water temperature was 25 ˚C. After 24 and 48 h, the dead larvae were counted, being considered as such, all those unable to reach the surface.

**Statistical analysis.** The experiment was carried out in triplicate. The larval mortality efficiency data were calculated in percentages using the Abbott formula and later tabulated in Microsoft Excel (Version 2013 for Windows). Probit analysis was performed with determination of the $LC_{50}$ (lethal concentration causing 50% mortality in the population) and the $LC_{90}$ (lethal concentration causing 90% mortality in the population) which were analyzed with a 95% confidence interval using the Statgraphics software Centurion XV version 15.2.11. The results were shown in the table. Differences that presented probability levels $p \leq 0.001$ for 24 h and $p \leq 0.013$ were considered statistically significant.

## Antimicrobial activity

**Microorganisms.** The antimicrobial EO test obtained from *O. majorana* L. leaves was tested in vitro against two gram-negative bacteria (*P. aeruginosa* ATCC 25922 and *E. coli* ATCC 8789) and a gram-positive bacteria (*S. aureus* ATCC 25922).

For each microorganism, the stock culture was stored in BHI medium (Brain Heart Infusion) with 20% glycerol and stored at –80 ˚C. An aliquot of 50 µL of this culture was inoculated into 5 mL of sterile BHI broth medium and incubated for 24 h at 37 ˚C.

**Determination of minimum inhibitory concentration (MIC) and minimum bactericidal concentration (MBC).** The MIC and MBC were determined using the microplate dilution technique (96 wells) according to the protocol established by Clinical and Laboratory Standards Institute [34], with adaptations.

Bacteria were initially reactivated from the stock cultures, kept in BHI broth, for 18 h at 37 ˚C. Subsequently, bacterial growth was prepared in 0.9% saline inoculum for each microorganism, adjusted to the McFarland 0.5 scale, then diluted in BHI and tested at 2 x 106 UFC. mL$^{-1}$ concentration.

In determining the MIC, the EO was diluted in Dimethylsulfoxide (2% DMSO). Each well of the plate was initially filled with 0.1 mL of 0.9% NaCl, except for the first column, which was filled with 0.2 mL of the EO at the concentration of 2000 µg.mL$^{-1}$. Subsequently, base two serial dilutions were performed in the ratio of 1:2 to 1:128 dilution in a final volume of 0.1 mL. After this process, 0.1 mL of cells (2 x 106 CFU mL$^{-1}$) added in each well related to the second preceding item, resulting in a final volume of 0.2 mL. Control of culture medium, control of EO, and negative control (DMSO 2%) were performed. And for the positive control, amoxicillin (0.5 µg.mL$^{-1}$) was used. After incubation of the microplates in an incubator at 37 ˚C for 24 hours, the plates were read in ELISA reader (OD 630nm).

The determination of MBC was performed based on the results obtained in the MIC test. Microplate wells were replicated in Müller-Hinton agar and incubated at 37 °C for 24 h. MBC was established as the lowest concentration of EO capable of completely inhibiting microbial growth.

**Statistical analysis.**   All experiments were performed in triplicate with the respective results categorized in Microsoft Excel (Version 2013 for Windows) and later analyzed in GraphPad Prism software (Version 6.0 for Windows, San Diego California USA). Significant differences between the groups were verified using the One-way ANOVA test with Bonferroni post-test. The data were considered statistically significant when p <0.001.

## Antioxidant activity

The antioxidant quantitative test was based on the methodology recommended by Sousa et al. [35], Lopes-Lutz et al. [36] and Andrade et al. [37] by the use of 2.2-diphenyl-1-picryl-hydra-zila (DPPH) with adaptations.

A methanolic solution of DPPH (stock solution) was prepared at the concentration of 40 $\mu g.mL^{-1}$, which was kept under the light. The EOs were diluted in methanol at concentrations 7.81; 15.62; 31.25; 62.5; 125 and 250 $\mu g.mL^{-1}$. For the evaluation of the test, 0.3 mL of the oil solution was added to a test tube, followed by the addition of 2.7 mL of the DPPH solution. White was prepared from a mixture with 2.7 mL of methanol and 0.3 mL of the methanol solution of each EO concentration as measured. After 30 min the readings were performed on a spectrophotometer (Biospectro SP-22) at a wavelength of 517 nm. The test was performed in triplicate and the calculation of the percentage of antioxidant activity (% AA) was calculated with the following Eq 1:

$$(\%AA) = 100 - \left\{ \frac{[(Abs_{sample} - Abs_{white}) \cdot 100]}{Abs_{control}} \right\} \tag{1}$$

AA%—Percentage of antioxidant activity
Abs_sample—Sample absorbance
Abs_white—White absorbance
Abs_control—Control absorbance

## Cytotoxic activity against *A. salina* L.

The cytotoxicity assay against *A. salina* L. Leach was based on the technique of Araújo et al. [38] and Lôbo et al. [39] with adaptations. An aqueous solution of artificial sea salt was prepared (35 $g.L^{-1}$) at pH 9.0 for incubation of 45 mg of *A. salina* L. eggs, which were placed in the dark for 24 h for the larvae to hatch (nauplii), then the nauplii were exposed to artificial light in 24 h, period to reach the stage methanuplii. The stock solution was prepared to contain 0.06 g of EO, 28.5 mL of solution of synthetic and 1.5 mL of Tween 80 to facilitate solubilization of the same. The test tubes were marked up to 5 mL. For the negative control, it was used respectively Tween 80 with solution saline (5%) and the ($K_2Cr_2O_7$) Potassium dichromate (1%) as the positive control.

The methanauplia were selected and divided into 7 groups of 10 subjects in each test tube. Each group received aliquots of the stock solution 100, 75, 50, 25, and 2.5 $\mu L$, which were then filled to a volume of 5 mL with the sea salt solution to produce final solutions with the following concentrations 40, 30, 20, 10, and 1 $\mu g.mL^{-1}$. The tests were performed in triplicates. For the test control, saline solution was used. After 24 h, the number of dead was counted.

**Statistical analysis.** The results obtained from the bioassays were expressed through Averages and Standard Deviation, categorized in Microsoft Excel (Version 2010 for Windows, Redmond, WA, USA). Significant differences between treatments were assessed using the ANOVA test One criterion and the Tukey test using the BioEstat program (Version 5.0 for Windows, Belem, BRA). The graphs were built on GraphPad Prism software (Version 6.0 for Windows, San Diego, CA, USA). The $LC_{50}$ values were determined in the PROBIT regression, through the SPSS statistical program (version 21.0 for Windows, Chicago, IL, USA). Differences that presented probability levels less than or equal to 5% ($p \leq 0.05$) were considered statistically significant.

## Results and discussion

### Identification of chemical compounds by GC-MS of the *O. majorana* L. EO

This result corroborates with other studies that have shown that environmental factors may affect certain chemical compounds, while in others they have no influence on their production [40, 41]. The chemical composition was determined by GC-MS, where the chromatogram can be observed in Fig 1.

On the chemical composition of the EO of *O. majorana* L. (Table 1), 95.8% are oxygenated monoterpenes and only 1.38% are monoterpene hydrocarbons. The major component of EO is pulegone (57.05%), followed by other components verbenone (16.92%), trans-menthone (8.57%), cis-menthone (5.58%), piperitone (2.83%), 3-octanol %) and isopulegol (1.47%).

Lima et al. [42], reports that piperitone has three organic functions in its chemical structure and it can be used for the synthesis of other compounds. Piperitone is derived from the metabolic pathway for the formation of piperitenone oxide, in which cis-pulegone is also, derived [43]. Macêdo et al. [44] observe that the variations of the active components of the plant are important parameters to correlate the activities, such as antibacterial and insecticide.

In addition, a number of biotic factors such as plant/ microorganism Stoppacher et al. [45], plants/insects Kessler and Baldwin [46] plant interactions, age and stage of development. As well as abiotic factors such as luminosity Takshak and Agrawal [47], temperature, precipitation, nutrition, time and harvest time Bitu et al. [48], they may present correlations with each other, acting together, and they may exert a joint influence on chemical variability and yield of essential oil [48].

### Larvicidal activity of EO of the *O. majorana* L. against *A. aegypti* L. larvae

The results of the larvicidal activity of this study show that *O. majorana* L. EO is active against *A. aegypti* L. larvae.

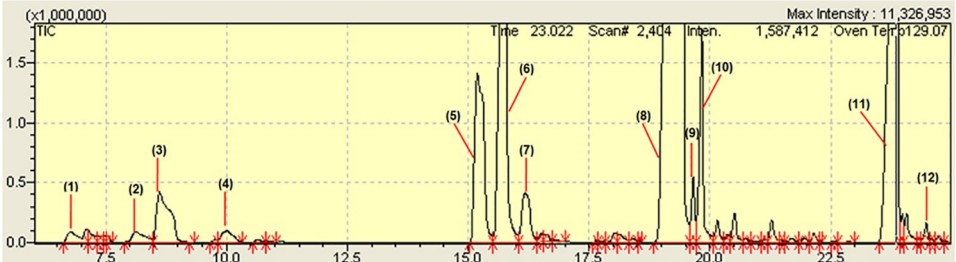

**Fig 1. Obtaining gas chromatography of OE *O. majorana* L. gas trap: Helium (He); initial temperature 60 ˚C; initial time 1.0 min; the column temperature increased 3 ˚C/ min. at 240 ˚C, maintained at this temperature for 30 min.**

**Table 1. Chemical composition of *O. majorana* L. essential oil.**

| N° * | IR | IK | Compounds | Relative Percentage (%) | Identification # |
|---|---|---|---|---|---|
| 1 | 6.776 | 939 | α-pinene | 0.39 | MS, IK |
| 2 | 8.127 | 979 | β-pinene | 0.50 | MS IK |
| **3** | **8.617** | **991** | **3-octanol** | **2.35** | **MS, IK** |
| 4 | 9.995 | 1029 | limonene | 0.49 | MS, IK |
| **5** | **15.185** | **1162** | **iso-menthone** | **5.58** | **MS, IK** |
| **6** | **15.688** | **1199** | **Trans-*p*-menthan-2-one** | **8.57** | **MS, IK** |
| **7** | **16.174** | **1149** | **isopulegol** | **1.47** | **MS, KI** |
| **8** | **19.448** | **1237** | **pulegone** | **57.05** | **MS, IK** |
| 9 | 19.655 | 1165 | lavandulol | 0.77 | MS, IK |
| **10** | **19.883** | **1252** | **piperitone** | **2.83** | **MS, IK** |
| **11** | **23.873** | **1205** | **verbenone** | **16.92** | **MS, IK** |
| 12 | 24.482 | 1161 | nonen-1-al-(2E) | 0.26 | MS, IK |
| | | | Total | 97.18 | |

*The identification path of the compounds,

#the identification of the compounds was performed by the mass spectrum (GC-MS) of the Library softwary Labsolutions GC-MS solution version 2.50 SU1 (NIST05 and WILEY'S libraries of the 9th edition mass spectrum); Kovats Index (KI) [32].

A fact that Komalamisra et al. [49], Magalhães et al. [50] and Dias et al. [51], classified with the values of the minimum lethal concentration that eliminates 50% of the population ($LC_{50}$) as a criterion for the activity. Because if $LC_{50} < 50$ µg.mL$^{-1}$, the product is considered very active, if $50 < LC_{50} < 100$ µg.mL$^{-1}$ the product is considered active, and when $LC_{50} > 750$ µg.mL$^{-1}$ the product is considered inactive.

The percentage of dead *A. aegypti* L. larvae is shown in Table 2, at different EO concentrations of *O. majorana* L. in the 24–48 h exposure period. There was no mortality in the control group. Through the probit test, $LC_{50} = 62.81$ µg.mL$^{-1}$, determination coefficient ($R^2$) = 78.64 and quantitative evaluation ($X^2$) = 24.238 in 24 h. After 48 h at $LC_{50} = 4.84$ µg.mL$^{-1}$, $X^2 = 10.3872$ and $R^2 = 55.16$.

The probit analysis performed was not visible the slope values, but as the P of the model in Table 3 demonstrated a statistically significant relationship between the variables at the 95% confidence level in 24 h, thus, the adjusted percentage of 78,64% was more appropriate and more effective than in 48 h, allowing inferring that mortality was slower with increased concentration.

**Table 2. Percentage of dead larvae (%) of *A. aegypti* L. produced by different concentrations of *O. majorana* L. essential oil in 24–48 h.**

| Concentrations | Larvicidal Activity (%) | |
|---|---|---|
| (µg.mL$^{-1}$) | 24 h | 48 h |
| Control (-) | 0.0 | 0.0 |
| 20 | 16 | 57.33 |
| 40 | 40 | 76 |
| 60 | 40 | 77.33 |
| 80 | 65.32 | 82.66 |
| 100 | 78.62 | 94.66[a] |
| $LC_{50}$ (positive control) | 0.0034 µg.mL$^{-1}$ | 0.0034 µg.mL$^{-1}$ |

[a] Statistically significant in relation to the positive control.

**Table 3. Insecticide response-concentration on larvae of *A. aegypti*.**

| Time | $CL_{50}$ ($IC_{95}$) µg.mL$^{-1}$ | $CL_{90}$ ($IC_{95}$) µg.mL$^{-1}$ | GL | $X^2$ | $R^2$ |
|------|------------------------------------|------------------------------------|-----|--------|--------|
| 24 h | 62.81 (50.87; 75.63) | 124.17 (103.23; 171.63) | 1 | 24.238 | 78.64 |
| 48 h | 4.84 (NI; 27.95) | 90.30 (70.16; 159.27) | 1 | 10.3872 | 55.16 |

NI: not identified; CI confidence interval; $X^2$; GL: Degree of Freedom; p <0.0001 for 24 h; p <0.0013 for 48 h.

There were no reports of studies on the larvicidal activity of *O. majorana* L. essential oil against *A. aegypti* L. larvae.

According to Cantrell et al. [52], larvicidal compounds act by absorption through the cuticle, via the respiratory tract, and/or enter by ingestion via the gastrointestinal tract. Once inside the larva, the substances may reach the site of action or may cause systemic effects by diffusion in different tissues [53].

Studies on the insecticidal effect of *Mentha* spp. reported that menthol, mentona, pulegone and carvone help to clarify the mechanisms of action on insects [54]. Previous studies indicate that limonene, camphene, and verbenone have insecticidal insect activity [55].

The effect of the tested essential oil of *O. majorana* L. can be explained by its chemical composition [56, 57]. The main components of essential oil, which belongs to the Lamiaceae family, are monoterpenes. In this context, the literature reports the larvicidal effect of monoterpene against mosquitoes [58, 59].

In the study by Pavela et al. [60] the essential oils of different species of *Mentha* L. and *Pulegium* showed varied chemical composition of monoterpenes, such as, for example, the major compound piperitone, which was effective and responsible for the insecticidal effects. Koliopoulos et al. [61] tested piperitenone oxide for larvicidal efficacy against the biotype *Culex pipiens* molestus and $LC_{50}$ estimated at 9.95 mg/l.

In general terms, it can be considered that the main entrance to the channel in the larvae of mosquitoes exposed to larvicides is the cuticular. In this process, the polarity (expressed as an octanol-water partition coefficient) of the xenobiotic that entry through the integument plays an essential role in determining the rate of entry into the body [62]

The results found in the literature on the chemical composition of the essential oil of *O. majorana* L. differ with the results of the species of this study collected in Macapá. It is worth mentioning that the plant species have several very different types of chemo and, at times, the participation of the main constituent may be only minor in essential oil.

Recently, a study by El-Akhal et al. [63], demonstrated the insecticidal activity of *O. majorana* L. against *Culex pipiens*, the vector of the West Nile virus. $LC_{50}$ and $LC_{90}$. The obtained LC were 258.71µg/mL and 580.49 µg/mL, respectively.

The $LC_{50}$ (107.13 µg/mL) and $LC_{90}$ (365.90 µg/mL) of the essential oil of *O. majorana* L., found in the study by El-Kahal et al. [63] against *An. Labranchiae*, are significant when compared to those found in *T. vulgaris*. They are also very important compared to those found in *O. majorana* L. against *Culex pipiens* in research work carried out in the laboratory [59]. The non-significant difference between the essential oils tested: *T. vulgaris* and *O. majorana* L. ($CI_{95}$ overlap) can be explained by their similar chemical composition [59, 63].

In fact, the main components of the two essential oils, which belong to the same family (Lamiaceae), are monoterpenes. In this context, the literature reports the larvicidal effect of monoterpene against mosquitoes [56].

Pavela and Sedlák [64] showed in their studies that insecticidal efficacy increased with temperature when OE was applied against *S. littoralis* larvae and revealed by a comparison of lethal

doses, while $LC_{50}$ and $LC_{90}$ for *S. littoralis* larvae in 15° C were estimated at 52 and 84 µg/larva, respectively, $LC_{50}$ and $LC_{90}$ were significantly lower at 30° C (32 and 56 µg/larva, respectively).

Despite the insecticidal efficacy of OEs has been studied in many insect species, very little information is available on other limiting factors of the post-application configuration that can have a significant impact on the insecticidal efficacy of OE-based botanical insecticides. Understanding the relationship between post-application temperature and insecticidal efficacy of OE is important, particularly in terms of practical recommendations for applying botanical insecticides based on EOs [64].

That is why this study tests the insecticidal efficacy of an essential oil obtained from *Thymus vulgaris* L. (Lamiaceae), applied at different ambient temperatures. *T. vulgaris* E was chosen intentionally because, compared to other OEs, it provides significantly better insecticidal efficacy and, therefore, was considered a highly promising active ingredient for some potential and already manufactured BIs [65].

However, when applied in water against *C. quinquefasciatus* larvae, the opposite effect was found. The water at the lowest temperature reached the highest mortality rate of *C. quinquefasciatus* and $LC_{50}$ (90) larvae were estimated at 21.3 (29.3) µg.L$^{-1}$ at 15 °C, while $LC_{50}$ (90) estimated 21.4 (33.1) µg.L$^{-1}$ were obtained for the highest temperature tested (30° C). This suggests that the influence of post-application temperature on insect mortality varies not only by the method of application, but also on the chemical composition of the OE [64].

Some EOs are known to cause dissuasive or anti-eating behavior in insects suggesting a neurotoxic action Satyan et al. [66], while some act as growth-regulating insects through analogous effects or antagonistic endogenous hormones. In the present study, it was found that even short-term exposure of larvae to lethal doses can dramatically increase their mortality over time and thereby reduce the total number of viable adults, leading to a possible reduction in total populations [67].

## Microbiological activity of *O. majorana* L. EO

In relation to the microbiological activity, it was possible to verify that gram-negative bacteria were more sensitive to *O. majorana* L. EO, than gram-positive bacteria.

The minimum inhibitory concentration (MIC) and minimum bactericidal concentration (MBC) that were identified for *O. majorana* L. EO can be verified in Fig 2. The results show that gram-negative bacteria were more sensitive to EO presenting MIC = 31.25 µg.mL$^{-1}$, compared to the negative control. The MBC for *E. coli* was at the concentration of 500 µg.mL$^{-1}$ and for *P. aeruginona* was at the concentration of 1000 µg.mL$^{-1}$ in relation to the negative control (amoxicillin). While the *S. aureus* bacterium did not present MIC, neither MBC.

According to Rosato et al. [68], the antibacterial activity in gram-negative bacteria occurs due to the high percentage of oxygenated monoterpenes present in the EO and consequently the synergism between these components. On the other hand, bacteria can also respond to adverse conditions in a transient way, through so-called stress tolerance responses. Bacterial stress tolerance responses include structural and physiological modifications in the cell, and complex genetic regulatory machines mediate them [69].

In the study by Duru et al. [70] pulegone showed high antimicrobial activity, particularly against *Candida albicans* and *Salmonella typhimurium*. Pulegone is classified as a monoterpene, in the same way as carvone. It can be obtained from a variety of plants [71, 72]. Menthone is a common volatile compound in Lamiaceae, which may also be active against a large number of bacteria, such as *E. coli* and *Enterococcus faecalis* [73, 74]. Some studies have argued that monoterpenes can cross cell membranes and interact with intracellular sites critical for antibacterial activity [75].

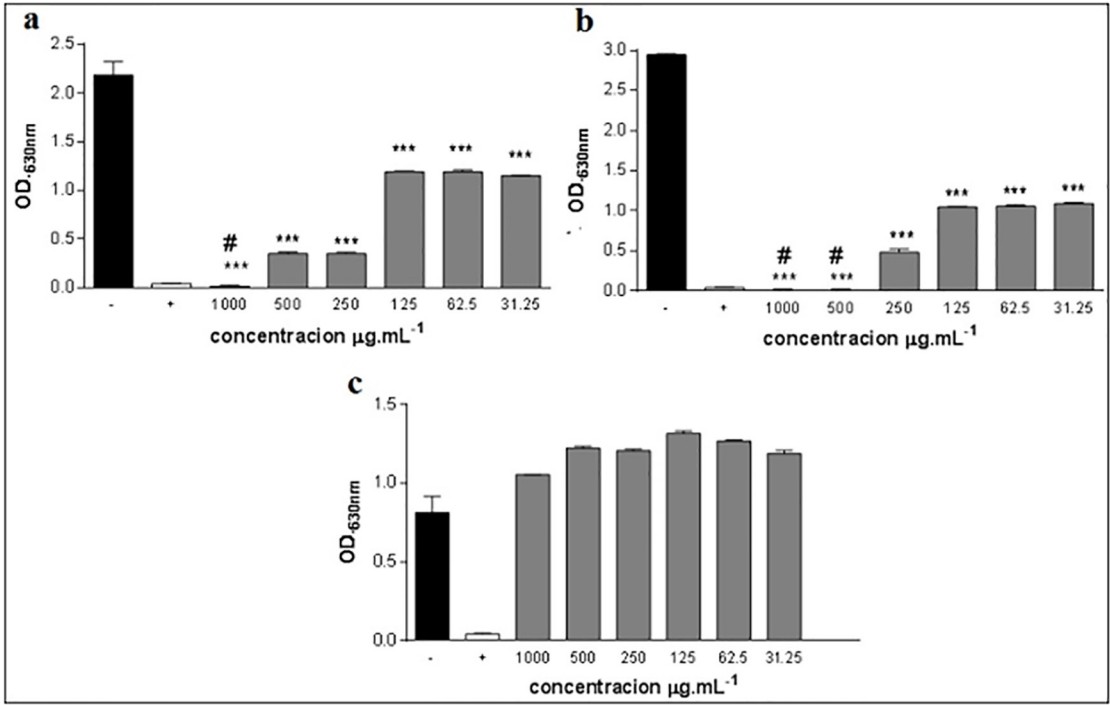

**Fig 2. This is the figure of Minimum Inhibitory Concentration (MIC) and Minimum Bactericidal Concentration (MBC) of *O. majorana* L. essential oil.** a) EO *O. majorana* L. against *P. aeruginosa* ATCC 25922. b) EO *O. majorana* L. against *E. coli* ATCC 8789. c) EO *O. majorana* L. against *S. aureus* ATCC 25922. Test substance of essential oil of the *O. majorana* L. (■), BHI with 2% of DMSO (■) and Amoxiline (□); ***P <0.001 statistically significant in relation to the negative control; # p <0.001 statistically significant in relation to the positive control.

However, reports of non-adaptation or cross-adaptation of bacteria to sub lethal concentrations of major constituents of essential oils has also been reported [76]. Cross-resistance can occur when different antimicrobial agents attack the same target in the cell, reach common route of access to the respective targets or initiate a common pathway for cell death, the resistance mechanism is the same for more than one antibacterial agent [77].

## Essential antioxidant activity of EO of *O. majorana* L. by DPPH radical capture method

The concentrations of EO obtained $IC_{50}$ = 16.83 μg.mL$^{-1}$ according to Table 4 and was compared with ascorbic acid (vitamin C) in Table 5 which showed $IC_{50}$ = 16.71 μg.mL$^{-1}$. The absorbance of EO was Y = 0.0196 x 17.0078. coefficient ($R^2$) = 0.9600.

**Table 4. Shows the percentage of the antioxidant activity of *O. majorana* L. EO.**

| Concentrations (μg.mL$^{-1}$) | AA (%) |
|---|---|
| 7.81 | 16.95[a a] |
| 15.62 | 17.15[b] |
| 31.25 | 17.48[b] |
| 62.5 | 18.9[c] |
| 125 | 19.28[d] |
| 250 | 21.85[e] |

Different letters indicate that there was significant difference of Tunkey test (p≤0.05).

**Table 5. Shows the percentage of antioxidant activity of ascorbic acid (vitamin C) in different concentrations.**

| Concentrations ($\mu g.mL^{-1}$) | AA (%) |
|---|---|
| 7.81 | 18.57%[a] |
| 15.62 | 30%[b] |
| 31.25 | 99.93%[c] |
| 62.5 | 99.99%[d] |
| 125 | 99.99%[d] |
| 250 | 99.99%[d] |

Different letters indicate that there was a significant difference ($p \leq 0.05$).

Many antioxidants derived from natural products demonstrate neuroprotective activity in vitro and/or in vivo models such as flavonoid phenolic compounds [78].

The percentage of antioxidant activity of the essential oil showed a high $IC_{50}$ = 16.83 $\mu g.mL^{-1}$, whereas ascorbic acid presented 16.71 $\mu g.mL^{-1}$ [79]. According to Rodrigues [80] the higher the consumption of DPPH for a smaller sample will be its $IC_{50}$ and the greater its antioxidant capacity.

According to Beatović et al. [81], the antioxidant capacity of EO is related to its main compounds. However, this study did not present antioxidant activity. The importance concerning the performance of antioxidants depends on the factors types of free radicals formed; where and how these radicals are generated; analysis and methods for identifying damage, and ideal doses for protection [82].

## Toxicity to *A. salina* L. EO of *O. majorana* L.

*A. salina* L. is a microcrustacean used in fish feed, and it is widely used in toxicological studies because of the low cost and easy cultivation. Several studies have attempted to correlate toxicity on *A. salina* L. with antifungal, virucidal, antimicrobial, trypanosomicidal and parasiticidal activities. Lethality assays are performed in toxicological tests and the median lethal concentration ($LC_{50}$), which indicates death in half of a sample, can be obtained [83].

Table 6 shows the percentage of cytotoxic activity of *O. majorana* L. EO and the mean mortality readings after the 24 h period are expressed. The oil concentrations presented $LC_{50}$ of 172.6 $\mu g.mL^{-1}$, the coefficient of determination ($R^2$) of 0.883 and $X^2$ of 1.915.

According to Nguta et al. [84], both organic extracts and aqueous extracts with $LC_{50}$ values of less than 100 $\mu g.mL^{-1}$ show high toxicity, $LC_{50}$ between 100 and 500 $\mu g.mL^{-1}$ exhibited

**Table 6. Shows the percentage of cytotoxic activity of *O. majorana* L. essential oil at different concentrations.**

| Concentrations ($\mu g.mL^{-1}$) | Mortality (%) |
|---|---|
| Control negative | 0.0%a |
| 50 | 9.37%b |
| 100 | 48.48%c |
| 250 | 67.39%d |
| 500 | 75.40%e |
| 750 | 80.26%f |
| 1000 | 83.51%g |
| $LC_{50}$ ($K_2Cr_2O_7$) | 172.60 $\mu g.mL^{-1}$ |

Different letters indicate a significant difference between the concentrations ($p < 0.05$).

moderate toxicity, $LC_{50}$ between 500 and 1000 μg.mL$^{-1}$ presented low toxicity and $LC_{50}$ above 1000 μg.mL$^{-1}$ are considered to be non-toxic (non-toxic).

The lethal concentration of mortality against the *A. salina* L. larvae of this assay showed moderate cytotoxic activity. In order to evaluate the cytotoxicity of a given sample, it is possible to elucidate the cytotoxic effect of the cytotoxic mechanism and the mechanism of action of different compounds during their interaction with the tissues [85].

## Conclusions

The results of the present study demonstrated that OE obtained from dry leaves of *O. majorana* L. showed good larvicidal activity against *A. aegypti* L. larvae with mortality from the concentration of 20 μg.mL$^{-1}$ in 48 h. In relation to the chemical analysis, it presented a mixture of monoterpenes, with the major constituent being pulegone (57.05%), followed by the other constituents verbenone (16.92%), trans-menthone (8.57%), cis-menthone), piperitone (2.83%), 3-octanol (2.35%) and isopulegol (1.47%). The oil showed satisfactory antimicrobial activity against *P. aeruginosa* and *E. coli* bacteria. In addition, despite the lack of antioxidant activity by the DPPH radical capture method, the oil showed moderate cytotoxic activity against *A. salina* L. In summary, these results provide subsidies for future EO *O. majorana* L. studies in order to enhance the use of organic compounds with larvicidal activity against the *A. aegypti* L. mosquito, as well as the importance of the study of bioactive plant products that do not pollute the environment and that do not cause harm to human health.

## Supporting information

**S1 File. Mass spectrum of *O. majorana* essential oil, obtained by GC-MS as compared to the spectrum of the equipment library NIST05 e WILEY'S and Adams (2017).**
(DOCX)

## Acknowledgments

To the Laboratory of Microbiology (LEMA) of UNIFAP under the responsibility of Prof. Aldo Proietti Aparecido Júnior.

To the Pro-rector of Research and Post-graduation—PROPESPG. Federal University of Amapá UNIFAP.

To the Laboratory Adolpho Ducke under the responsibility of Eloise Andrade.

## Author Contributions

**Conceptualization:** Sheylla Susan Moreira da Silva de Almeida.

**Data curation:** Sheylla Susan Moreira da Silva de Almeida.

**Formal analysis:** Renata do Socorro Barbosa Chaves, Allan Kardec Ribeiro Galardo, Sheylla Susan Moreira da Silva de Almeida.

**Investigation:** Renata do Socorro Barbosa Chaves, Rosany Lopes Martins, Alex Bruno Lobato Rodrigues, Érica de Menezes Rabelo, Ana Luzia Ferreira Farias, Lethicia Barreto Brandão, Lizandra Lima Santos, Allan Kardec Ribeiro Galardo.

**Methodology:** Sheylla Susan Moreira da Silva de Almeida.

**Project administration:** Sheylla Susan Moreira da Silva de Almeida.

**Resources:** Sheylla Susan Moreira da Silva de Almeida.

**Supervision:** Sheylla Susan Moreira da Silva de Almeida.

**Writing – original draft:** Renata do Socorro Barbosa Chaves, Sheylla Susan Moreira da Silva de Almeida.

**Writing – review & editing:** Sheylla Susan Moreira da Silva de Almeida.

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
