## [Decision Letter · Decision Letter 0]

8 Jan 2020

PONE-D-19-23621

Evaluation of larvicidal potential against larvae of Aedes aegypti (Linnaeus, 1762) and of the antimicrobial activity of essential oil obtained from the leaves of Origanum majorana L.

PLOS ONE

Dear Dr de Almeida,

Thank you for submitting your manuscript to PLOS ONE. After careful consideration, we feel that it has merit but does not fully meet PLOS ONE’s publication criteria as it currently stands. Therefore, we invite you to submit a revised version of the manuscript that addresses the points raised during the review process.

The reviewers of your study require a significant change in the manuscript as detailed below. Please provide a letter addressing all of the comments raised by the reviewers.

We would appreciate receiving your revised manuscript by Feb 22 2020 11:59PM. To enhance the reproducibility of your results, we recommend that if applicable you deposit your laboratory protocols in protocols.io, where a protocol can be assigned its own identifier (DOI) such that it can be cited independently in the future. For instructions see: http://journals.plos.org/plosone/s/submission-guidelines#loc-laboratory-protocols

We look forward to receiving your revised manuscript.

Kind regards,

Horacio Bach

Academic Editor

PLOS ONE

Journal Requirements:

https://peerj.com/articles/5476.pdf

In your revision ensure you cite all your sources (including your own works), and quote or rephrase any duplicated text outside the methods section. Further consideration is dependent on these concerns being addressed.

3. Thank you for stating the following financial disclosure: "No"

a. Please provide an amended Funding Statement that declares *all* the funding or sources of support received during this specific study (whether external or internal to your organization) as detailed online in our guide for authors at http://journals.plos.org/plosone/s/submit-now

b. Please state what role the funders took in the study.  If any authors received a salary from any of your funders, please state which authors and which funder. If the funders had no role, please state: "The funders had no role in study design, data collection and analysis, decision to publish, or preparation of the manuscript."

4. Please amend the manuscript submission data (via Edit Submission) to include authors Camila Mendes da Conceição Vieira Araújo and Talita Fernandes Sobral.

"To the Amapá Foundation for Research Support (FAPEAP). To the Research Program of SUS - PPSUS - Ministry of Health. To the Coordination of Improvement of Higher Education Personnel (CAPES)/Ministry of Education (MEC). To the National Council of Scientific and Technological Development - CNPQ. To the Laboratory of Microbiology (LEMA) of UNIFAP under the responsibility of Prof. Aldo Proietti Aparecido Júnior. To the Pro-rector of research and post-graduation - PROPESPG. Federal University of Amapá UNIFAP. To the Laboratory Adolpho Ducke under the responsibility of Eloise Andrade."

"No"

6. We note you have included a table to which you do not refer in the text of your manuscript. Please ensure that you refer to Table 3 in your text; if accepted, production will need this reference to link the reader to the Table.

Reviewers' comments:

Reviewer's Responses to Questions

**Comments to the Author**

1. Is the manuscript technically sound, and do the data support the conclusions?

Reviewer #1: Partly

Reviewer #2: No

2. Has the statistical analysis been performed appropriately and rigorously? 

Reviewer #1: Yes

Reviewer #2: No

3. Have the authors made all data underlying the findings in their manuscript fully available?

Reviewer #1: No

Reviewer #2: No

4. Is the manuscript presented in an intelligible fashion and written in standard English?

Reviewer #1: Yes

Reviewer #2: Yes

5. Review Comments to the Author

Reviewer #1: Dear authors,

This manuscript deals with the chemical composition of Origanum majorana essential

oil and its mosquitocidal, antimicrobial, cytotoxic and antioxidant potential. The results of this laboratory study are interesting in that they extend our knowledge about larvicidal potential against Aedes aegypti and antimicrobial activity of Origanum majorana essential oil.

Further down you will find some suggestions of major importance that would probably benefit the manuscript.

Line 126: In the statistical analysis for the larvicidal bioassays it should be noted that the dose response data were analyzed by probit analysis and that LC50, LC90 values along with the confidential limits and the slope were generated.

Line 255: In the results concerning the larvicidal effect against mosquitoes, I propose to add a table with the LC50 and LC90 values with the confidential limits, along with the slope values and x2 and R2, for the treatment with the essential oil and the positive control at 24 and 48 hours in order for the reader to compare statistically the performance of the treatments.

Line 105: As a major drawback of the study design is that the major compounds, that were identified in the essential oil used, were note tested against Aedes aegypti larvae. The performance of the major compounds would help to interpret the toxicity of the essential oil and identify toxic active substances against Aedes aeypti larvae. Please comment.

Lines 273-274: The discussion concerning the larvicidal effect of the essential oil should be elaborated more presenting literature data for the larvicidal effect of O. majorana essential oil against other mosquito species.

Lines 279-282: The larvicidal effect of the major compounds of the essential oil used, namely pulegone, verbenone, trans-menthone, cis-menthone, piperitone, 3-octanol and isopulegol against primarily Aedes aegypti and other mosquito species from the literature should be discussed, in order to interpret the action of the current essential oil against Aedes aegypti.

Reviewer #2: The authors studied the larvicidal, antimicrobial, cytotoxic and

antioxidant potential of Origanum majorana essential oil.

Although the topic is relatively interesting and is suitable for publication in PONE, the MS shows some defects that prevent me from recommending the MS for publication.

Major comments:

1. In the introduction, explain goals of the work better. Why did you choose plant - Origanum majorana and why EO and not need extracts - in terms of industrial use of plants. What is the practical impact of work? How does antioxidant capacity and bactericidal activity relate to insecticidal activity? Why other non-target organisms have not been selected.

2. Results: The authors give lethal doses; however, if they want to find correct and relevant values, they should use the probit analysis and compare the CI95 intervals. Only in this way can they claim that a significant difference or progression in efficacy in time has been achieved. The term Positive Control (Table 2) is not correct. This term is used to describe the control variant where a commercial insecticide has been used (to compare biological efficacy of a standard insecticide and EOs). It is not clear if the percentages (eg Table 5) were transformed before statistical analysis. The calculation of LD50 (Table 2) for 48 hours is inaccurate because the lowest mortality was 57%, and for 24 hours the mortality for the two different concentrations was the same (40%) - it is not clear which of the values was taken to calculate the lethal concentration.

3. The discussion should be improved. Chemical composition is very important for the discussion and for understanding the mechanism of action. The authors should give a better comparison of their achieved efficacy to other authors who studied EOs with a majority content of pulegone (see e.g. DOI: 10.17221/48/2013-PPS; https://doi.org/10.1016/j.indcrop.2015.06.050 ect.) or EOs obtained from Origanum majorana against mosquitoes (see e.g. DOI: 10.12980/APJTB.4.2014APJTB-2014-0392; Int J Pharm Pharm Sci, Vol 8, Issue 3, 372-376 ect.). EO efficacy can be influenced by various external factors including the post-application temperature – this should be discussed (see e.g. DOI: 10.1016/j.indcrop.2018.01.021). The efficacy of sublethal doses or concentrations has been studied – and this should be discussed, as well. The implementation method is yet another aspect to be discussed.

6. PLOS authors have the option to publish the peer review history of their article (what does this mean?). If published, this will include your full peer review and any attached files.

Reviewer #1: No

Reviewer #2: No

---

## [Author Response · Author response to Decision Letter 0]

29 Feb 2020

Response to Revisores

Reviewer #1: Dear authors,

This manuscript deals with the chemical composition of Origanum majorana essential oil and its mosquitocidal, antimicrobial, cytotoxic and antioxidant potential. The results of this laboratory study are interesting in that they extend our knowledge about larvicidal potential against Aedes aegypti and antimicrobial activity of Origanum majorana essential oil. Further down you will find some suggestions of major importance that would probably benefit the manuscript.

Line 126: In the statistical analysis for the larvicidal bioassays it should be noted that the dose response data were analyzed by probit analysis and that LC50, LC90 values along with the confidential limits and the slope were generated.

The requirement was met, as can be seen in the lines 158-162 and 163.

Line 255: In the results concerning the larvicidal effect against mosquitoes, I propose to add a table with the LC50 and LC90 values with the confidential limits, along with the slope values and x2 and R2, for the treatment with the essential oil and the positive control at 24 and 48 hours in order for the reader to compare statistically the performance of the treatments.

The requirement was met, as can be seen in the lines 305-312.

Line 105: As a major drawback of the study design is that the major compounds, that were identified in the essential oil used, were note tested against Aedes aegypti larvae. The performance of the major compounds would help to interpret the toxicity of the essential oil and identify toxic active substances against Aedes aeypti larvae. Please comment.

The requirement was met, as can be seen in the lines 132-134.

Lines 273-274: The discussion concerning the larvicidal effect of the essential oil should be elaborated more presenting literature data for the larvicidal effect of O. majorana essential oil against other mosquito species.

The requirement was met, as can be seen in the lines 315 – 328.

Lines 279-282: The larvicidal effect of the major compounds of the essential oil used, namely pulegone, verbenone, trans-menthone, cis-menthone, piperitone, 3-octanol and isopulegol against primarily Aedes aegypti and other mosquito species from the literature should be discussed, in order to interpret the action of the current essential oil against Aedes aegypti.

The requirement was met, as can be seen in the lines 293 – 302.

Reviewer #2: The authors studied the larvicidal, antimicrobial, cytotoxic and antioxidant potential of Origanum majorana essential oil. Although the topic is relatively interesting and is suitable for publication in PONE, the MS shows some defects that prevent me from recommending the MS for publication. Major comments:

1. In the introduction, explain goals of the work better. Why did you choose plant - Origanum majorana and why EO and not need extracts - in terms of industrial use of plants. What is the practical impact of work? How does antioxidant capacity and bactericidal activity relate to insecticidal activity? Why other non-target organisms have not been selected.

The requirement was met, as can be seen in the lines has 53–61 and 86-93.

2. Results: The authors give lethal doses; however, if they want to find correct and relevant values, they should use the probit analysis and compare the CI95 intervals. Only in this way can they claim that a significant difference or progression in efficacy in time has been achieved.

The requirement was met, as can be seen in the lines -310-317.

The term Positive Control (Table 2) is not correct. This term is used to describe the control variant where a commercial insecticide has been used (to compare biological efficacy of a standard insecticide and EOs).

The requirement was met, as can be seen in the lines 308 - 309.

It is not clear if the percentages (eg Table 5) were transformed before statistical analysis.

No. The percentages were calculated before statistical analysis. Statistical analyzes were performed based on percentages.

The calculation of LC50 (Table 2) for 48 hours is inaccurate because the lowest mortality was 57%, and for 24 hours the mortality for the two different concentrations was the same (40%) - it is not clear which of the values was taken to calculate the lethal concentration.

The statistical analysis was redone to understand the data more precisely. The requirement has been met, as can be seen on lines 303–305. The two 40% concentrations in Table 2 remain the same in 24 hours, as they are part of the larvae mortality control, but do not compromise the OE larvicidal activity, being more active in 24 h than in 48 h. Mortality decreased dramatically during 48 hours and in the static analysis it presented estimated values with very low and insignificant variation in larval mortality, but this result does not compromise the effective larvicidal activity of the EO.

3. The discussion should be improved. Chemical composition is very important for the discussion and for understanding the mechanism of action. The authors should give a better comparison of their achieved efficacy to other authors who studied EOs with a majority content of pulegone (see e.g. DOI: 10.17221/48/2013 PPS; https://doi.org/10.1016/j.indcrop.2015.06.050 ect.) or EOs obtained from Origanum majorana against mosquitoes (see e.g. DOI: 10.12980/APJTB.4.2014APJTB-2014-0392; Int J Pharm Pharm Sci, Vol 8, Issue 3, 372-376 ect.)

The requirement was met, as can be seen in the lines 323-335 e 341–353.

EO efficacy can be influenced by various external factors including the post-application temperature – this should be discussed (see e.g. DOI: 10.1016/j.indcrop.2018.01.021). The efficacy of sublethal doses or concentrations has been studied – and this should be discussed, as well. The implementation method is yet another aspect to be discussed.

The requirement was met, as can be seen in the lines -354-377.

Tables:

Line 429: Table 3 changed to Table 4

Line 435: Table 4 changed to Table 5

Line 4561: Table 4 changed to Table 6

References

Line 511 correction

3. Gonçalves NVS, Rebêlo JMM. [Epidemiological aspects of dengue in the municipality of São Luís, Maranhão, Brazil, 1997-2002]. Public Health Notebooks. 2004; 20: 1424-1431. Doi: 10.1590/S0102-311X2004000500039.

Line 516 correction

4. Barbosa IR, Araújo LF, Carlota FC, Araújo RS, Maciel IJ. [Epidemiology of dengue fever in the State of Rio Grande do Norte, Brazil, 2000 to 2009]. Epidemiol Serv Health 2012; 21: 149-57. Doi:V10.5123/S1679-49742012000100015. 

Line 524: correction

6. Vasconcelos PFC. Zika virus disease: a new emerging problem in the Americas? Rev. PanAmaz. Saúde. 2015; 6: 9-10. Doi: 10.5123/S2176-62232015000200001. 

Line 856: correction

78. Santos JMP. [Adaptation and Cross-Adaptation of Listeria Spp. Essential Oils from Condensed Plants and Acid Stress]. [dissertation]. Postgraduate Program in Medicinal, Aromatic and Spice Plants: University of Lavras; 2018.

Line 874: correction

82. Rodrigues JSQ. [Infusions based on leaves of passifloras of the cerrado: phenolic compounds, antioxidant activity in vitro and sensorial profile]. [dissertation] Universidade de Brasília. 2012. 

Line 629: correction

31. Anvisa. National Health Surveillance Agency. [Brazilian Pharmacopoeia], 5a ed.; Fiocruz: Brasília, Brasil 2010, pp. 1-545. Português.

Line 655: correction.

37. Andrade MA, Cardoso GM, Batista RL, Mallet TCA, Machado FMS. [Essential Oils of Cymbopogon Nardus, Cinnamomum Zeylanicum and Zingiber Officinale: composition, antioxidant and antibacterial activities]. J Agron Sci. 2012; 43 (2): 399-408. Doi: 10.1590/S1806-66902012000200025. Português.

Line 532-567 Added in References.

8. Zara, AL.; Santos, SM. dos; Oliveira-Fernandes, ES.; Carvalho, RG.; Coelho, GE. [Strategies for controlling Aedes aegypti: a review]. Epidemiol Serv Health. 2016; 25 (2): 391-404. Doi: 10.5123/s1679-49742016000200017. 

9. Araújo VEM, Bezerra JMT, Frederico FA, Passos VMA, Carneiro M. [Increased burden of dengue in Brazil and federated units, 2000 and 2015: analysis of the Global Burden of Disease Study 2015]. Brazilian Journal of Epidemiology. 2017; 20: 205-216. Doi: 10.1590/1980-5497201700050017. 

10. Wang X, Li JL ,Xing HJ, Xu SW. Review of the toxicology of atrazine and chlorpyrifos onfish. Journal Northeast Agric Univ, 2011, (18): 88-92. 

11. Mohammed MP, Penmethsa KK. Assessment of pesticide residues in surface waters of Godavari delta, India. Journal Mater Environ Sci, 2014, (5): 33-6.

12. Djogbénou L. Vector controls methods against malaria and vector resistance to insecticides in Africa. Med Trop, 2009, (69): 160-4. 

13. El Ouali Lalami A, El-Akhal F, El Amri N, Maniar S, Faraj C. State resistance of the mosquito Culex pipiens towards temephos central Morocco. Bull Soc Pathol Exot Ses Fil, 2014, (107): 194-8.

14. Azizi A, Yan F, Honermeier B. Herbage yield, essential oil content and composition of three oregano (Origanum vulgare L.) populations as affected by soil moisture regimes and nitrogen supply. Ind Crops Prod 2009, (29): 554-61.

15. Abdel-Massih RM, Abraham A. Extracts of Rosmarinus officinalis, Rheum rhaponticum, and Origanum majorana exhibit significant anti-staphylococcal activity, International Journal of Pharmaceutical Sciences and Research, 2014, (5): 819– 828.

16. Busatta C, Vidal RS, Popiolski AS. Application of Origanum majorana L. essential oil as an antimicrobial agent in sausage. Food Microbiology. 2008, 25 (1): 207–211.

17. Prakash B, Singh P, Kedia A, Dubey NK. Assessment of some essential oils as food preservatives based on antifungal, antiaflatoxin, antioxidant activities and in vivo efficacy in food system. Food Research International, 2012, 49.(1): 201–208.

Line 588-616 Added in References.

22. Dhaheri YAl, Attoub S, Arafat K. Anti-metastatic and anti-tumor growth effects of Origanum majorana on highly metastatic human breast cancer cells: inhibition of NFκB signaling and reduction of nitric oxide production. PLoS ONE, 2013, 8, (7) Article IDe68808.

23. Rao S, Timsina B, Nadumane VK. Evaluation of the anticancer potentials of Origanum marjorana on fibrosarcoma (HT-1080) cell line. Asian Pacific Journal of Tropical Disease, 2014, 4 (1): S389–S394.

24. Ramadan G, El-Beih NM, Zahra MM. Egyptian sweet marjoram leaves protect against genotoxicity, immunosuppression and other complications induced by cyclophosphamide in albino rats. British Journal of Nutrition, 2012, 108,.(6): 1059– 1068.

25..Khan JA, Jalal J.A, Ioanndes C,. Moselhy SS. Impact of aqueous doash extract on urinary mutagenicity in rats exposed to heterocyclic amines. Toxicology and Industrial Health, 2013, 29, (2): 142–148.

26. V´agi E, Sim´andi B, Suhajda´ Aethelyi´ EH. Essential oil composition and antimicrobial activity of Origanum majorana L. extracts obtained with ethyl alcohol and supercritical carbon dioxide, Food Research International, 2005, 38 (1): 51–57.

27. Ebrahimi, M, Khosravi-Darani, K, Essential oils as natural food preservatives: antimicrobial and antioxidant applications. In: Doughari, James (Ed.), Antimicrobials from Nature: Effective Control Agents for Drug Resistant Pathogens. 2013, Transworld Research Network, Kerala, (2): 37-661, ISBN: 978-81-7895-603-9

28. Bassalé, JRN., Juliani, H.R., (2012). Essencial oils in combination and their antimicrobial properties. Molecules, 3989-4006.

30. Pereira ÁIS, Pereira AGS, Sobrinho OPL, Cantanhede EKP, Siqueira LFS. [Antimicrobial activity in the control of larvae of the mosquito Aedes aegypti: Homogenization of the essential oils of linalool and eugenol]. Chemical education 2014; 25 (4): 446-449. Doi: 10.1016/S0187-893X(14)70065-5. 

Line 754-805 Added in References.

56. El-Akhal F, Greche H, Ouazzani CF, Guemmouh R, El Ouali Lalami A. Chemical composition and larvicidal activity of Culex pipiens essential oil of Thymus vulgaris grown in Morocco. JMater Environ Sci 2015;1:214-9.

57. El-Akhal F, El Ouali Lalami A, Ez Zoubi Y, Greche H, Guemmouh R. Chemical composition and larvicidal activity of essential oil of Origanum majorana (Lamiaceae) cultivated in Morocco against Culex pipiens (Diptera: Culicidae). Asian Pac J Trop Biomed2014;4:746-50.

58. Szczepanik M, Zawitowska B, Szumny A. Insecticidal activities of Thymus vulgaris essential oil and its components (thymol and carvacrol) against larvae of lesser mealworm, Alphitobius diaperinus Panzer (Coleoptera Tenebrionidae). Allelopathy J2012;30:129-42.

59. Cavalcanti ESB, de Morais SM, Lima MAA, Santana EWP. Larvicidal activity of essential oils from Brazilian plants against Aedes aegypti L. Mem Inst Oswaldo Cruz 2004;99:54 1-4. 

60. Pavela R, Kaffková K, Kumsta M. Chemical composition and larvicidal activity of essential oils from different Mentha L. and Pulegium species against Culex quinquefasciatus Say (Diptera: Culicidae). Plant Protect. Sci. 2014; (50): 36–42. DOI: 10.17221/48/2013-PPS

61. Koliopoulos G., Pitarokili D., Kioulos E., Michaelakis A., Tzakou O. (2010): Chemical composition and larvicidal evaluation of Mentha, Salvia and Melissa essential oils against the West Nile virus mosquito Culex pipiens. Parasitology Research, 107: 327-375.

62. Lucia A, Zerba E, Masuh H. Knockdown and larvicidal activity of six monoterpenes against Aedes aegypti (Diptera: Culicidae) and their structure-activity relationships. Parasitol Research. 2013, (112): 4267-4272. DOI: 10.1007/s00436-013-3618-6.

63. El-Akhal F, Guemmouh R, Maniar S, Taghzouti K, Lalami A E O, Larvicidal Activity of Essential Oils of Thymus Vulgaris and Origanum Majorana (Lamiaceae) Against of the Malaria Vector Anopheles Labranchiae (Diptera: Culicidae). 2016, International Journal of Pharmacy and Pharmaceutical Sciences. (8); 372-376, issue 3, DOI: 10,12980 / APJTB.4.2014APJTB-2014-0392

64. Cavalcanti ESB, de Morais SM, Lima MAA, Santana EWP. Larvicidal activity of essential oils from Brazilian plantsagainst Aedes aegypti L. Mem Inst Oswaldo Cruz2004;99:541-4. 

65. Pavela R, Sedlák P. Post-application temperature as a factor influencing the insecticidal activity of essential oil from Thymus vulgaris. Industrial Crops e Products. 2018, (113); 46-49. DOI: 10.1016 / j.indcrop.2018.01.021

66. Pavela R. History, presence and perspective of using plant extracts as commercial botanical insecticides and farm products for protection against insects –a review. Plant Prot. Sci. 2016, (52): 229–241.

67. Dias CN, Moraes DFC. Essential oils and their compounds as Aedes aegypti L. (Diptera: Culicidae) larvicides: review. Parasitology Research. 2014, 113 (1): 565- 592.

---

## [Decision Letter · Decision Letter 1]

23 Mar 2020

PONE-D-19-23621R1

Evaluation of larvicidal potential against larvae of Aedes aegypti (Linnaeus, 1762) and of the antimicrobial activity of essential oil obtained from the leaves of Origanum majorana L.

PLOS ONE

Dear Dr de Almeida,

Thank you for submitting your manuscript to PLOS ONE. After careful consideration, we feel that it has merit but does not fully meet PLOS ONE’s publication criteria as it currently stands. Therefore, we invite you to submit a revised version of the manuscript that addresses the points raised by Reviewer #1 during the review process.

We would appreciate receiving your revised manuscript by May 07 2020 11:59PM. To enhance the reproducibility of your results, we recommend that if applicable you deposit your laboratory protocols in protocols.io, where a protocol can be assigned its own identifier (DOI) such that it can be cited independently in the future. For instructions see: http://journals.plos.org/plosone/s/submission-guidelines#loc-laboratory-protocols

We look forward to receiving your revised manuscript.

Kind regards,

Horacio Bach

Academic Editor

PLOS ONE

Reviewers' comments:

Reviewer's Responses to Questions

**Comments to the Author**

1. If the authors have adequately addressed your comments raised in a previous round of review and you feel that this manuscript is now acceptable for publication, you may indicate that here to bypass the “Comments to the Author” section, enter your conflict of interest statement in the “Confidential to Editor” section, and submit your "Accept" recommendation.

Reviewer #1: (No Response)

Reviewer #2: All comments have been addressed

2. Is the manuscript technically sound, and do the data support the conclusions?

Reviewer #1: Yes

Reviewer #2: Yes

3. Has the statistical analysis been performed appropriately and rigorously? 

Reviewer #1: Yes

Reviewer #2: Yes

4. Have the authors made all data underlying the findings in their manuscript fully available?

Reviewer #1: Yes

Reviewer #2: Yes

5. Is the manuscript presented in an intelligible fashion and written in standard English?

Reviewer #1: Yes

Reviewer #2: Yes

6. Review Comments to the Author

Reviewer #1: Dear authors,

Thank you for considering my comments that have been addressed in the manuscript noting however the following:

In table 2 it should be clarified if the LC50 value for the positive control was at 24 or 48 hours. Also, the significant difference in the larvicidal effect between the tested EO and the positive control should be mentioned in the text since the positive control was included in the study.

Please note that the chemical analysis gives us an indication of the implication of the components to the effect of the essential oil, however the most appropriate approach is to test the major compounds against the target organism.

I recommend to remove Lines 357-373 from the final (clean) draft manuscript (That is why this study tests the insecticidal….composition of the OE [66]) as non-relevant detailed information about the effect of temperature in insecticidal effect, considering that this was not subjected in the current research work

Reviewer #2: Authors corrected the manuscript according to reviewers' comments, I have no further comments.

7. PLOS authors have the option to publish the peer review history of their article (what does this mean?). If published, this will include your full peer review and any attached files.

Reviewer #1: No

Reviewer #2: No

---

## [Author Response · Author response to Decision Letter 1]

30 Apr 2020

Response to Reviewers

1. Please amend your list of authors on the manuscript to ensure that each author is linked to an affiliation.

We note that you have included affiliation symbols '&' and '¶' beside author names, however there are no affiliations linked to these symbols. 

Please amend to link an affiliation 4 to each or remove if added in error.

The requirement was met, as can be seen in the lines 5-8.

Renata do Socorro Barbosa Chaves1, Rosany Lopes Martins1, Alex Bruno Lobato Rodrigues1, Érica de Menezes Rabelo1, Ana Luzia Ferreira Farias1, Lethicia Barreto Brandão1, Lizandra Lima Santos1, Camila Mendes da Conceição Vieira Araújo2, Talita Fernandes Sobral2, Allan Kardec Ribeiro Galardo2, Sheylla Susan Moreira da Silva de Almeida1*

2. Thank you for including your ethics statement: 'N/A' and thank you for stating in your methods section that:

'The leaves of O. majorana L. were collected in the district of Fazendinha (00 "36'955" S and 51 "11'03'77" W) in the Municipality of Macapá, Amapá. Five samples of the plant species were deposited at the Amapaense Herbarium (HAMAB) of the Institute of Scientific Research and Technology of Amapá (IEPA).'

To comply with PLOS ONE submissions requirements for field studies, please provide the following information in the Methods section of the manuscript and in the “Ethics Statement” field of the submission form (via “Edit Submission”):

a) Provide the name of the authority who issued the permission for each location (for example, the authority responsible for a national park or other protected area of land or sea, the relevant regulatory body concerned with protection of wildlife, etc.). If the study was carried out on private land, please confirm that the owner of the land gave permission to conduct the study on this site.

The requirement was met, as can be seen in the lines 103.

a) The study was carried out on private land with permission from the owner.

b) For any locations/activities for which specific permission was not required, please

- i. state clearly that no specific permissions were required for these locations/activities, and provide details on why this is the case

- ii. confirm that the field studies did not involve endangered or protected species

The requirement was met, as can be seen in the lines 104-106.

i. There was no need for specific permissions for the study site/activity, as it is a private property where the owner has given authorization to collect the plant material.

ii. The study did not involve threatened or endangered species.

---

## [Decision Letter · Decision Letter 2]

4 May 2020

PONE-D-19-23621R2

Evaluation of larvicidal potential against larvae of Aedes aegypti (Linnaeus, 1762) and of the antimicrobial activity of essential oil obtained from the leaves of Origanum majorana L.

PLOS ONE

Dear Dr de Almeida,

Thank you for submitting your manuscript to PLOS ONE. After careful consideration, we feel that it has merit but does not fully meet PLOS ONE’s publication criteria as it currently stands. Therefore, we invite you to submit a revised version of the manuscript that addresses the points raised during the review process.

Please address the new comment addressed by the reviewer. 

We would appreciate receiving your revised manuscript by Jun 18 2020 11:59PM. To enhance the reproducibility of your results, we recommend that if applicable you deposit your laboratory protocols in protocols.io, where a protocol can be assigned its own identifier (DOI) such that it can be cited independently in the future. For instructions see: http://journals.plos.org/plosone/s/submission-guidelines#loc-laboratory-protocols

We look forward to receiving your revised manuscript.

Kind regards,

Horacio Bach

Academic Editor

PLOS ONE

Reviewers' comments:

Reviewer's Responses to Questions

**Comments to the Author**

1. If the authors have adequately addressed your comments raised in a previous round of review and you feel that this manuscript is now acceptable for publication, you may indicate that here to bypass the “Comments to the Author” section, enter your conflict of interest statement in the “Confidential to Editor” section, and submit your "Accept" recommendation.

Reviewer #1: (No Response)

2. Is the manuscript technically sound, and do the data support the conclusions?

Reviewer #1: Yes

3. Has the statistical analysis been performed appropriately and rigorously? 

Reviewer #1: Yes

4. Have the authors made all data underlying the findings in their manuscript fully available?

Reviewer #1: Yes

5. Is the manuscript presented in an intelligible fashion and written in standard English?

Reviewer #1: Yes

6. Review Comments to the Author

Reviewer #1: Dear authors,

My comments were partially addressed in the manuscript not affecting, however, the robustness of the study. As a final comment, please check if the LC50 values for the positive control (esbiothrin) against mosquito larvae were indeed the same for 24 and 48 records of mortality.

7. PLOS authors have the option to publish the peer review history of their article (what does this mean?). If published, this will include your full peer review and any attached files.

Reviewer #1: No

---

## [Author Response · Author response to Decision Letter 2]

12 Jun 2020

Response to reviewers

Comments to the Author

6. Review Comments to the Author

Reviewer #1: Dear authors,

My comments were partially addressed in the manuscript not affecting, however, the robustness of the study. As a final comment, please check if the LC50 values for the positive control (esbiothrin) against mosquito larvae were indeed the same for 24 and 48 records of mortality.

The requirement was met, as can be seen in the lines 298-300 and in the table 305-306.

Dear reviewer, analyzing Table (in the lines 312-313), it was necessary to use bold in the title of table 3; remove the spacing that existed between the title and table 3; add the parenthesis after the LC50 value 124.17 (103.23; 171.63) in 24 h, apply the superscript and subscript of the LC50 values (IC95) µg.mL-1, CL90 (IC95) µg.mL-1, R2, X2 and apply italic in X2. In addition, it was necessary to indent the paragraph after table 3 (in the lines 314-315).

Regarding the corrections in the references, the word "Portuguese" was removed on lines 357, 625, 656, 857, 874, 892 and 905.

---

## [Decision Letter · Decision Letter 3]

23 Jun 2020

Evaluation of larvicidal potential against larvae of Aedes aegypti (Linnaeus, 1762) and of the antimicrobial activity of essential oil obtained from the leaves of Origanum majorana L.

PONE-D-19-23621R3

Dear Dr. de Almeida,

We’re pleased to inform you that your manuscript has been judged scientifically suitable for publication and will be formally accepted for publication once it meets all outstanding technical requirements.

Kind regards,

Horacio Bach

Academic Editor

PLOS ONE

Additional Editor Comments (optional):

Reviewers' comments:

Reviewer's Responses to Questions

**Comments to the Author**

1. If the authors have adequately addressed your comments raised in a previous round of review and you feel that this manuscript is now acceptable for publication, you may indicate that here to bypass the “Comments to the Author” section, enter your conflict of interest statement in the “Confidential to Editor” section, and submit your "Accept" recommendation.

Reviewer #1: All comments have been addressed

2. Is the manuscript technically sound, and do the data support the conclusions?

Reviewer #1: Yes

3. Has the statistical analysis been performed appropriately and rigorously? 

Reviewer #1: Yes

4. Have the authors made all data underlying the findings in their manuscript fully available?

Reviewer #1: Yes

5. Is the manuscript presented in an intelligible fashion and written in standard English?

Reviewer #1: Yes

6. Review Comments to the Author

Reviewer #1: Dear authors

Thank you for considering my comments. After your last response the manuscript could be accepted for publication.

7. PLOS authors have the option to publish the peer review history of their article (what does this mean?). If published, this will include your full peer review and any attached files.

Reviewer #1: No

---

## [Editor Report · Acceptance letter]

30 Jun 2020

PONE-D-19-23621R3 

Evaluation of larvicidal potential against larvae of Aedes aegypti (Linnaeus, 1762) and of the antimicrobial activity of essential oil obtained from the leaves of Origanum majorana L. 

Dear Dr. de Almeida:

I'm pleased to inform you that your manuscript has been deemed suitable for publication in PLOS ONE. Congratulations! Your manuscript is now with our production department. 

Kind regards, 

on behalf of

Dr. Horacio Bach 

Academic Editor

PLOS ONE